# The Versatility of Polymeric Materials as Self-Healing Agents for Various Types of Applications: A Review

**DOI:** 10.3390/polym13081194

**Published:** 2021-04-07

**Authors:** Nik Nur Farisha Nik Md Noordin Kahar, Azlin Fazlina Osman, Eid Alosime, Najihah Arsat, Nurul Aida Mohammad Azman, Agusril Syamsir, Zarina Itam, Zuratul Ain Abdul Hamid

**Affiliations:** 1School of Materials & Mineral Resources Engineering, Engineering Campus, Universiti Sains Malaysia, Nibong Tebal 14300, Malaysia; niknur.farisha8@gmail.com (N.N.F.N.M.N.K.); najihaharsat@gmail.com (N.A.); 2Faculty of Chemical Engineering Technology, University Malaysia Perlis (UniMAP), Arau 02600, Malaysia; azlin@unimap.edu.my; 3Biomedical and Nanotechnology Research Group, Center of Excellence Geopolymer and Green Technology (CEGeoGTech), Universiti Malaysia Perlis (UniMAP), Arau 02600, Malaysia; 4King Abdulaziz City for Science and Technology (KACST), P.O. Box 6086, Riyadh 11442, Saudi Arabia; alosimi@kacst.edu.sa; 5Department of Civil Engineering, Universiti Tenaga Nasional, Selangor 43000, Malaysia; aidam386@gmail.com; 6Institute of Energy Infrastructure, Universiti Tenaga Nasional, Selangor 43000, Malaysia; agusril@uniten.edu.my

**Keywords:** self-healing polymeric materials, self-healing mechanism, intrinsic and extrinsic healing mechanisms

## Abstract

The versatility of polymeric materials as healing agents to prevent any structure failure and their ability to restore their initial mechanical properties has attracted interest from many researchers. Various applications of the self-healing polymeric materials are explored in this paper. The mechanism of self-healing, which includes the extrinsic and intrinsic approaches for each of the applications, is examined. The extrinsic mechanism involves the introduction of external healing agents such as microcapsules and vascular networks into the system. Meanwhile, the intrinsic mechanism refers to the inherent reversibility of the molecular interaction of the polymer matrix, which is triggered by the external stimuli. Both self-healing mechanisms have shown a significant impact on the cracked properties of the damaged sites. This paper also presents the different types of self-healing polymeric materials applied in various applications, which include electronics, coating, aerospace, medicals, and construction fields. It is expected that this review gives a significantly broader idea of self-healing polymeric materials and their healing mechanisms in various types of applications.

## 1. Introduction

Failure is a potential occurrence in any human invention such as buildings, spaceships, electronic devices, and medical devices. All these products or structures are susceptible to damage due to many factors, one being environmental factors, such as environmental threats like mechanical stresses, chemical abrasion, ultra-violet irradiations, thermal loading, and pressure. Generally, materials tend to be damaged at the micro- or macrolevel. At the microlevel, cracking for instance, affects the material’s characteristics, while at the macrolevel, serious damage can occur from cracking as it can cause the material or structure to destruct, which, in turn, leads to more catastrophic impact [1].

Take for example a concrete structure subjected to microcracking due to the weakness of concrete in tension. Fatigue loading can reduce the concrete’s toughness, which leads to the increment of permeability, and hence, can ultimately lead to the reduction of the concrete’s structural integrity, durability, and life span. In medical application, microcracking, wear and tear, fatigue, and material deformation and loss often arise from cyclic loading or aggressive usage, which can cause mechanical failure in biomedical implants [2]. Surface scratching and microcracking in the coating system may promote the formation of cracks, which eventually leads to macroscopic damage, up to the point where the coating loses its aesthetic as well as its external protective function. Exposure to a high electric field may also have the potential to cause mechanical damage in dielectric and insulating elements, which can lead to the failure of electronic devices and electrical equipment.

Other examples of major material failure modes that are routinely encountered in composite materials in aerospace application are fiber-matrix debonding, matrix microcracking, and impact damage. These examples of damage initiation may lead to more serious damage or can also contribute to catastrophic failure. These catastrophic failures may cause significant impacts on repair, maintenance, and the rehabilitation of existing and new structures. Apparently, deterioration of products or structure is an ongoing and concerning problem. The consequential effects of damage may have far reaching implications if the damage is not controlled at an early stage. Thus, this catastrophic failure can have a significant influence on the economic, social, and environmental impacts. Therefore, preliminary action should be addressed at the microlevel to prevent serious damage from happening.

Self-healing methods, which refer to the ability of a system to recover after initial damage, have the potentials to address these challenges. Self-healing technology can mitigate the molecular level functionality, enabling the repair, impart reprocess ability, or longer lifetimes of products. The concept of self-healing was inspired from natural or biological process, for example, blood clotting or reparation of fractured bones. This concept was then incorporated at a wider range of engineering applications. In some cases, manual repair works can be difficult, especially if the cracks or damages are invisible, or their locations cannot be easily accessed.

Therefore, self-healing process provides the best solution because the system has the capability to repair itself as per the biological system, without the need for any human detection or repair by manual intervention. One of the strategies to address the microcrack or inaccessible damage sites is by using self-healing polymers. This is because self-healing polymers have high efficiency to recover the properties of the components on three basis stages; which are (1) triggering, (2) the healing material is transported to the damage site, and (3) remodeling by the chemical repair process [3]. With the application of the self-healing method in each system, the lifetime of the material may be extended, which in turn can reduce the cost of maintenance and improve product safety.

Self-healing technologies were progressively developed in various kinds of applications. Market research has portrayed the increase of self-healing products at a value of USD $381.65 M in 2019 and is expected to stretch to USD $943.38 M by 2027. The building industry segment held approximately 56.5% of the market, as this industry is the largest consumer for self-healing material. Other sectors, such as automotive and transportation, energy generation, electronics and semiconductors, and the medical sector, have also contributed to the increasing of the self-healing material market [4].

In developing self-healing technology, many innovations have been explored including research from the National University of Singapore (NUS) Institute for Health Innovation and Technology and NUS Materials Science and Engineering. They have developed a durable, low-power material for next-generation electronic wearables and soft robots. This unique material has very high dielectric permittivity and self-healing properties, which enables it to store more electronic charges at lower voltages and a higher brightness when used in a light-emitting capacitor device. The material is a transparent, elastic rubber sheet made up of a unique blend of Fluor elastomer and surfactant. This innovation is called the HELIOS (which stands for healable, low-field illuminating optoelectronic stretchable) device [5]. Another innovation developed at the University of Technology in the Netherlands is the self-healing “bio” concrete that uses nutrients (peptone and yeast extract) and Bacillus Subtilis without urea as a microbial adjuvant, and this could also be an effective way to promote microbial CaCO_3_ precipitation in sufficient quantities to enhance sustainability and to promote self-healing of concrete [6].

To date, the number of publications with various applications of self-healing materials has risen dramatically [7,8,9,10,11,12]. For instance, B. Aissa et al. summarized the main technologies that are currently being developed specifically in thermosetting composites polymeric system [12]. Besides that, Qiang et al. also reviewed on recently developed self-healing conductive materials and their applications [7]. Most of the recent reviews reported the emerging of self-healing materials in specific applications with their self-healing mechanism. A comprehensive review on the self-healing mechanisms, approaches, polymeric materials used, and their applications is still limited. Thus, due to the remarkable and emerging self-healing products to date, this review paper summarizes the self-healing polymer technologies in different types of applications, such as in electronics, coating, aerospace, the medical field, and construction. We discuss thoroughly on their principles, interaction mechanisms, self-healing polymeric materials used, and their respective effectiveness. The different types of self-healing polymers for various types of applications are explored. In the successive section, the healing process is classified and described according to the self-healing mechanism (embedded, vascular, and intrinsic polymer-based system). Finally, in order to gain an overall insight into the self-healing materials, the advantages and disadvantages of the self-healing approach in every application and the current challenges of self-healing technologies are also discussed. Therefore, this review paper is expected to significantly give a broader idea of self-healing polymers, which in turn proves to be the solution to increasing the life cycle of materials, and also concurrently advocates a safer environment in all the applications involved.

## 2. Self-Healing Polymer Principle and Mechanism

Self-healing is described as a method which has the ability to perceive any unfunctional device or system without human intervention and make the necessary adjustments to restore itself back to its normal condition. The self-healing method is expected to detect and repair the microcracking in products or system that help to recover the mechanical or functional performance of the material. It represents the next generation of technology, which helps to greatly improve important performance of products, as it has been built in into various applications for structural, electronic, medical and aerospace products [13,14,15,16]. This approach is very significant in situations where repairing or inspection is difficult, hazardous, and costly [17].

Over the last decade, self-healing methods have attracted a lot of attention in the research community and numerous studies have explored them in order to develop various self-healing capabilities, particularly using polymeric materials. Polymeric materials are formed by the bonding of monomer and polymer chains and they are one of the most extensively researched materials in self-healing because of their versatility in polymer systems. For instance, a self-healing process that requires relatively low temperature can be easily achieved by the modification and functionalization of polymer system and can effectively fill the cracks because of the higher viscosity of polymer. Besides that, the self-healing polymers also can transform physical energy into a chemical and/or physical response to heal the damage. Polymers are in demand as they offer a light weight material with good processability and chemical stability [18]. In particular, efficient self-healing polymers and polymer-based fiber-reinforced composites are of high significance because these materials are utilized in numerous engineering fields, such as for cars, ships, spacecraft, electronics, and biomedicine [19,20,21].

### 2.1. General Self-Healing Mechanism

There are two types of self-healing mechanisms which are the extrinsic and intrinsic self-healing mechanisms. Normally, nearly all self-healing mechanisms take place at room temperature. In the extrinsic mechanism, the healing agent embedded in the polymer matrix recovers the damage, whereas in the intrinsic mechanism, the chemical bonding of the matrix itself has the ability to restructure in existence of the external action [3].

In the extrinsic mechanism, the self-healing approach is via the presence of an intentionally added external healing agent, which can be encapsulate or in vascular network. As the damage occurs, the component, the capsule or the vascular network tends to break down and release the healing agents into the damaged regions. Furthermore, the healing agents and catalysts can be sequestered in four different ways: capsule-catalyst, multicausal, latent functionality, and phase-separated. The micro-capsulate diameter and surface morphology influence the behavior of the capsule rapture, and the healing agent is released [3].

However, the encapsulated method has certain disadvantages, as it allows for healing only once, with the capsule break and release of the healing agent. On the other hand, the vascular self-healing systems offer multiple healing reactions, where healing agents and catalysts are sequestered in networks in the form of capillaries or hollow channels. The restorative agent is embedded into a network vessels or hollow canals, which are pumped through a thin nanosized or micro sized of vascular tube network, and then the healing agent is released upon the appearance of microcracking. Throughout years of exploration, self-healing vascular systems have been improved from 2D to 3D interconnected networks, which enhances the self-healing agent flowability and increases the mechanical properties of the composite. Particularly, the vascular self-healing method is like the capsule-based method, defined as the need of a healing agent, but through different approaches. The healing mechanisms for the embedded and vascular methods are illustrated in Figure 1.

Many researchers have explored the development of encapsulated and vascular self-healing. Fernando et al. (2019) developed an epoxy self-healing coating by using encapsulated epoxy ester resin in poly(urea-formaldehyde-melamine) microcapsules. The filled microcapsules were incorporated into an epoxy three-layer coating system. The coating system containing the microcapsules showed a significant self-healing effect when stressed by a mechanical defect, and the higher concentration (15 wt%) of microcapsules provided better self-healing protection with better anticorrosive performance [22]. Naveen et al. (2020) focused on self-healing microcapsules encapsulated with carbon nanotubes for improved thermal and electrical properties. In this study, multiwalled nanotubes (CNTs) were successfully incorporated with the dicyclopentadiene (DCPD) core and encapsulated by an urea-formaldehyde (UF) shell, to form dual-core microcapsules. The development of dual-core microcapsules with the DCPD–CNT–UF combination was found to improve the mechanical, thermal, and electrical properties of the resin cast specimens without compromising on self-healing efficiency [23]. Li et al.’s (2013) studies showed that cementitious materials with microcapsules consist of healing adhesives. The adhesive materials, like diglycidyl ether of biphenyl and epoxy resin (E) and poly styrene-divinylbenzene (stn-DVB), were mixed with a cement paste to achieve self-healing capability. They demonstrated a significant improvement in the strength parameters [24].

Meanwhile, for the vascular network system, De Nardi et al. (2020) proposed mini-vascular networks (MVNs) as an alternative for the vascular method. The MVNs considered in this study were 3D-printed tetrahedral units (TETs) of interconnected hollow ligaments that had a characteristic dimension (i.e., maximum ligament length) approximately twice the size of the maximum aggregate particle size of the cementitious composite. A tetrahedral shape is also advantageous because the ligaments are straight between the anchor points (apices), which means the ligaments are more likely to break when crossed by a crack (see Figure 2) [25].

Compared to the extrinsic mechanism, the intrinsic self-healing mechanism does not need an external agent to repair the damage [3]. This process allows for multiple and repeatable self-repair after numerous of damage or healing [26]. Besides that, intrinsic self-healing polymer materials have various advantages like (i) fast self-healing of functions due to the absence of diffusion and polymerization control steps and (ii) adaption in restoring low-volume defects/cracks because of their sensitive reversible dynamic bonds in molecular scale [26]. Intrinsic self-healing can be achieved through the inherent reversibility of molecular interactions of the matrix polymer, which evades complex problems such as the integration and compatibility of healing agents. Materials with intrinsic healing properties are triggered by external stimuli, such as electrical or photo stimuli, heat or pressure, or the reaction of substances when introduced together, to prompt them to heal. For intrinsic self-healing, molecular mechanisms can occur from physical interactions and/or chemical interactions. The healing process is achieved by tailoring dynamic bond interactions from the molecular level or formulating interplay between the diffusion and entanglement of polymer chains [26]. Chemical bond interaction-based self-healing can be further classified into covalent and non-covalent bond interactions.

#### 2.1.1. Covalent Bond Interaction

The formation of chemical bonds between different molecules or functional groups of a polymer can be described by the reversibility of the covalent bond, where the opening and closing of the bonding sites results in an equilibrium which serves as the basis for the self-healing process [27]. Besides that, external conditions such as temperature, pressure or electrical stimuli can affect the chemical bonds. Generally, this mechanism can be accomplished with a high or low temperature. However, changes in the pH value, irradiation, moisture, or catalytic activity also can affect the disconnection and reconnection of the active sites [27]. The reversibility of the cross-links in the polymers provides a wide range of possible binding concepts with regards to the polymer structure itself. The network formation is held either in the main chain, side chain, or with a multifunctional linker, for example disulfide bonds, Diels–Alder chemistry, imine bonds, di-selenide bonds, boron-based bonds, transesterification reactions, ditelluride bonds, and others (see Figure 3).

#### 2.1.2. Non-Covalent Bond Interaction

Intrinsic self-healing mechanisms of the non-covalent nature comprise weak interactions that can occur between different families of atoms, such as van der Waals interactions, p–p stacking, dipole–dipole interactions, hydrogen bonding, ionic interactions, metal–ligand coordination, and host–guest interactions as well as materials associated by non-covalent bonds, i.e., supramolecular polymers or gels, which can self-heal with or without external stimuli, as shown in Figure 4. Because the non-covalent systems have lower bonding energy than the covalent systems, they typically have higher healing efficiencies, which allows for the restoration of the broken bonds, even at room temperature. The hydrogen bond is the most used in non-covalent interactions of elastomer, which have been combined with other interactions of the same nature to create self-healing materials with various performances [28].

A few studies on the intrinsic dynamic covalent and non-covalent healing mechanism have been reported. Chen et al. (2019) recently reported the self-healing method based on intrinsic dynamic non-covalent mechanism, where polyurethane (urea) elastomer is formed by van der Waals forces and hydrogen bonds. Self-healing polyurethane demonstrated that the dynamic cross-linking between hydrogen bonds and van der Waals forces is the basic driving force for the self-healing ability of the material, and temperature is the key factor that affects hydrogen bonding and van der Waals forces [29]. Elena et al. (2017) also reported on the self-healing non-covalent interaction by using diphenyl disulfide-based compounds. The self-healing properties of disulfide-based material are mainly created through the capacity to generate sulfinyl radicals by breaking the disulfide S-S bond, and the ability of these radicals to attack neighboring disulfides. Additionally, the hydrogen bond interactions are established between chains. The presence of hydrogen bonds in the repeating unit of the polymer plays a role in the crystallization of the material, stabilizing adjacent chains [30]. Kuhl et al. (2018) focused on the exchange reactions of the reversible urethane moieties. Reversible covalent urethane units were used for the design of new self-healing polymers, which had efficiencies of up to 85%, while the material exhibited good mechanical properties with high E-moduli [31].

## 3. Self-Healing Polymer in Electronic Applications

To date, electrical devices have been progressively developed because of the current challenging requirements and demands. They have been built to be smaller and some have become miniature in size, lighter, flexible, and portable [32]. Electronic devices have rapidly evolved into highly flexible, curvilinear formats that enable a new range of wearable applications. Compared with rigid electronic systems limited to planar configurations, soft electronic system designs can be stretched, compressed, bent, and deformed into arbitrary shapes without electrical or mechanical failure in circuits. Those soft applications have promoted a better interaction of the device with humans and also the environment (see Figure 5).

On the other hand, the soft electronic devices are not tough enough to withstand the unexpected mechanical damage caused by repeated wear and tear, accidental cutting or scratching, excess temperature, excess current or voltage, ionizing radiation, mechanical shock, or stress or impact, which are the main causes of device failure [32]. These failures promote microcracking or so-called electric treeing. Electrical degradation fundamentally has a different mechanism and damaging process from mechanical damage compared to other applications. In electrical devices, the damaging event always happens in dielectric polymer, which includes the electrical treeing process as well as electrochemical, photochemical, and thermochemical degradation, followed by the catastrophic breakdown. This process causes the formation of dendric hollow cracks of few micrometers in the tube diameter [33]. Usually, it is initiated by an insulation defect like impurities, gas voids, mechanical defects, or conducting projections that cause excessive electrical field stress within small regions of the dielectric.

Therefore, monitoring and repairing microcracking inside the device becomes difficult and almost impossible. Self-healing is a potential approach to overcome this problem. It has proven to be successful when it comes to integration into functional electronic devices including sensors, supercapacitors, perovskite solar cells, field-effect transistors (FETs), and batteries [34]. In advanced technologies, self-healing methods can also be applied in wearable electronics and artificial skin-like materials [35]. In electronic applications, there are only few numbers of self-healing polymeric systems that have been used because of the numerous requirements of such applications (including combining mechanical and electrical properties, electrical and environmental stability, and scalability) as well as a lack of understanding of the necessary design rules [32].

### 3.1. Self-Healing Mechanism in Electronics

The self-healing process of electronically active polymeric materials is triggered by dynamic equilibrium of the cross-linking network and percolation pathways in the polymeric system. It occurs by utilizing dynamic bonds such as secondary, covalent, and supramolecular bonds [36]. Moreover, the mobility of the polymer chains, the concentration of the broken dynamic bonds available, and the activation energies for the exchange of dynamic bonds determines the healing rate process. In certain cases, solvent vapor is also required to activate polymer chains and to cause them to rearrange. Because the polymeric materials have low conductivity, design strategies for creating self-healing and high-performance electronic materials have primarily focused on incorporating electronically active fillers into a dynamic polymer matrix. It was recently demonstrated that nanostructured conductive networks based on one-dimensional wires, in contact with a cross-linked self-healing polymer matrix, can be dynamically reorganized to allow for the retrieval of the original percolation paths for electrical conduction after mechanical damage. Moreover, the kinetics of self-healing and the electrical responses can be affected by the size and surface properties of the conductive filler in a polymer matrix [37]. The concept for the electronically self-healing materials is shown in Figure 6.

In the extrinsic mechanism, self-healing is enabled within composite systems using functional particles such as microcapsules filled with liquids, micro/nanodroplets, and solid fillers. Similarly, in the electronic device, when a composite system is damaged, microcapsules release healing agents, microdroplets rupture and coalesce, forming new connections, and solids reestablish contact at time scales dependent on the network mobility. When a crack occurs in the line, the microcapsules break open and release a liquid metal that fills the crack and restores the flow of electricity. In addition to restoring the current within microseconds, the microcapsules also demonstrated outstanding reliability with 90% of the samples restoring up to 99% of the original conductivity, even with a small number of microcapsules being used. Meanwhile, in the intrinsic mechanism for electronic devices, self-healing materials are commonly fabricated by dynamic covalent bonds (for instance, Diels-Alder reaction or disulfide bonds) or non-covalent bonds (for example, hydrogen bonds, π-π interactions, metal-ligand interactions, host-guest complexes, and ion–dipole interactions), and intermolecular interactions, which can spontaneously heal the damage and recover mechanical properties, and thus, restore the conductivity [35].

### 3.2. Self-Healing Polymeric Materials and Their Efficiency as Healing Agents in Electronics

The use of self-healing polymers in electrical applications has shown significant impact as the developed technologies have promoted self-repair inside the devices without the need of monitoring. Yan et al. (2020) reported an application of self-healing materials in smart batteries and supercapacitors [38]. In this application, the most widely used polymers were conductive polymer gels (CPGs). CPGs have been used to create multifunctional mixed materials and have been discovered to repair damage at the fracture interface via reversible chemical bonds or specific interactions (for example, ligand–metal bonding and hydrogen bonding). The high conductivity and flexibility of conductive hydrogels have made them functional materials for several applications such as bioelectrodes, biosensors, electronic-skin, and implantable biorobots. Common polymers used in such electronic devices are polyvinylidene fluoride (PVDF), sodium alginate (SA), carboxymethyl cellulose (CMC), and polyacrylic acid (PAA). These polymers are used as binders and their healing ability is determined by the cycling performance of the self-healing Si electrode [38].

In the advance electronic applications, self-healing also can be applied in stretchable conductors for soft electronics. Stretchable electronics are devices that consist of electronic materials and/or circuits integrated onto stretchable substrates. Compared to rigid printed circuit boards, stretchable electronic circuits can mechanically bend, twist, compress, and stretch. This is because they use elastomeric soft substrate materials. For example, conductive polymers (e.g., poly(3,4-ethylenedioxythiophene): poly(styrene sulfonate) (PEDOT:PSS), polyaniline (PANi), and polypyrrole (PPy)) are used as healing agents [36]; polypyrrole and ferric ions self-healing is accomplished by using a polymer substrate capable of a reversible Diels–Alder (DA) reaction [39]. Ryuzu et al. (2020) introduced a stretchable and conductive self-healing elastomer based on intermolecular networks formed by poly (acrylic acid) (PAA) and reduced graphene oxide (rGO) through a facile and suitable post reduction and one-pot method presented in their study. The rGO and PAA-GO elastomers have stretchable and conductive characteristics which have good mechanical stability and electrical properties. Furthermore, by constructing covalent (for example, ester bonds) and non-covalent (for instance, hydrogen bonding) intermolecular networks between poly (acrylic acid) (PAA) and rGO, these materials demonstrate both electrical and mechanical self-healing properties. After cutting, the elastomers self-healed quickly within 30 s and efficiently reached 95% recovery at room temperature. The elastomers were accurate and reliable in detecting external strain even after the healing process. The elastomers were also used for strain sensors that were directly attached to human skin to monitor external movements, as well as finger bending and wrist twisting [35].

Moreover, stretchable polymer chain that has been incorporated with conductive material can form a self-healable cross-linked network. The polymers used included poly (vinylidene fluoride-co-hexafluoropropylene), poly(3,4-ethylenedioxythiophene) polystyrene sulfonate (PEDOT:PSS), low-density polyethylene (LDPE), cross-linked polyethylene (XLPE) polypropylene (PP), polyimide (PI), epoxy, polydimethylsiloxane (PDMS), and polyvinylidene fluoride (PVDF) [33]. The above-mentioned polymers interacted with the ionic liquid through strong ion–dipole interaction to form a self-healable cross-linked network, which exhibits reasonable ionic conductivity. Nonetheless, Yang et al. (2020) stated that failure of electronic devices and electrical equipment classically occurs in the dielectric and insulation elements that are essential to withstand strong electric fields and/or supportive conductors of high electric potential. The high demand for electricity coupled with the widespread usage of consumer electronics has increased the demand for advanced dielectric polymers with substantially improved reliability and lifespan. As a result, dielectric and insulating polymers with self-healing functionalities are becoming an interesting subject of research study to address arising challenges [33]. Table 1 tabulates a further example of self-healing polymeric materials in electronics.

Self-healing polymer material has a good potential to bring about a better future for electronic devices because it increases the lifespan of electronics and also reduces waste. However, patterning self-healing materials with the small feature sizes required for integrated electronics applications remains challenging and time consuming. Thus, the development of photo-patternable or self-healing electronic materials is required. Besides that, the availability of functional materials for applications remains limited, especially semiconductors and conductors. Furthermore, the electronic properties of the current self-healable devices are not effective as compared to the non-healable functional electronic devices. New design concepts for self-healing electronic polymers or the hybridization of self-healing polymers with high-performance inorganic nanomaterials is necessary to improve their electronic properties [33].

## 4. Self-Healing Polymers in Coating Application

Recently, self-healing polymers and polymer composites have been used widely in the application of protective coating, especially to protect metallic materials from corrosion [40]. It is not only low cost, but the polymer-based coating provides excellent protection against corrosive ions and is applied in various sectors and industries [41]. Generally, the self-polymer coating works by restoring the protective layer of coating barriers by defect sealing or by preventing further corrosion that occurs in the damaged coating.

### 4.1. Self-Healing Mechanism in Coatings

There are two types of self-healing mechanism for polymer-based coatings: (1) Autonomous healing, in which polymerizable healing agents or other additives such as corrosion inhibitors are embedded in the matrix of the polymeric coating material. Autonomous corrosion protection is developed through regeneration of protective properties in the damaged coating area; (2) Non-autonomous healing, in which the healing mechanism is induced by external energy, such as heat, light stimuli, or organic solvent, that leads to further reactions, chemically or physically. As a result, molecular chain movement and bond formation can be developed for the healing process [42,43].

The main advantage of the autonomous self-healing coating is that it can prolong the coating’s lifetime, thereby bringing a positive impact to the economy and environment [44]. Furthermore, this type of coating is very useful to protect the corrosive parts that are located in difficult-to-access areas/environments such as in aerospace, small devices, the deep sea, and inside the human body [42]. In the autonomous self-healing coating polymers, there are two self-healing strategies that commonly involve: (i) the intrinsic method that employs various way of dynamic chemistries and non-covalent interactions and (ii) the extrinsic method through a capsule-based system (microencapsulation) [44,45]. Most intrinsic systems achieve healing through reversible covalent bonds (for example Diels–Alder reactions, imine bonds, and disulfide metathesis), reversible hydrogen bonding, metal–ligand bonding, or a combination of the two (for example, hydrogen bonding interactions with dynamic imine bonds and hydrogen bonding interactions with disulfide metathesis). Figure 7 summarizes the damage repair mechanisms in the polymer-based coating systems with their main applications.

The intrinsic self-healing system that involves reversible bonding can provide multiple cycles of healing effect. However, external factors are needed to induce healing, such as thermal treatment of the damaged part. In organic chemistry, the Diels–Alder (DA) reaction is important to produce reversible healing through its thermal reversibility. Thus, it is possible to develop self-healing polymers with well-defined structures and properties by using this reaction.

### 4.2. Self-Healing Polymeric Materials and Their Efficiency as Healing Agents in Coatings

Based on the DA reaction mentioned previously, Kotteritzsch et al. (2013) synthesized terpolymers containing functional moieties by using a DA reaction in order to produce reversible cross-linking that can generate the self-healing capability of a coating. The terpolymers are composed of methacrylate backbone containing furan and maleimide units in the side chains. They claimed that the self-healing property of the coating can be obtained even without the addition of the cross-linker. The damaged coating material can be heated to a temperature at which a retro-DA reaction takes place. When the coating is cooled to room temperature, the scratch/damage can be healed because of the coupling of the two reactive functional groups. Moreover, the self-healing process can occur repeatedly to produce a damage-free coating [45].

The extrinsic self-healing system usually involves the incorporation of capsules into a polymer matrix. Inside these microcapsules, there are healing agents that can be released to the targeted areas (damaged region) by capillary effects once the capsules rupture. However, when using this microencapsulated approach, the self-healing component must be of low viscosity, so that it can be stored in the microcapsules and released when needed. Furthermore, the capsules must be strong enough to be handled and dispersed in the polymers, yet can be easily breakable upon matrix cracking. The released healing agent can heal the damaged region of the matrix through physical or/and chemical interactions [46,47]. Jialan et al. (2019) used an in-situ polymerization process to prepare self-healing microcapsules using urea formaldehyde as the wall material and epoxy resin (E-51) as the core material. The changes in the particle size of the microcapsules were analyzed in relation to the amount of the epoxy resin, ammonium chloride, dodecylbenzene sulfonate (SDBS), and resorcinol used. Furthermore, the effect of the stirring rate on the particle size of the microcapsules was also investigated. They found the best formulation for the microcapsule was with 24% epoxy resin to urea ratio, 8% SDBS, 5.6% ammonium chloride, and 10.4% resorcinol. In addition, the optimum stirring speed was 450 rpm. By using these parameters, the obtained microcapsule size was 55.7 µm. The microcapsules were then applied into the epoxy composite coating for the evaluation of corrosion resistance. The coating could achieve the best corrosion resistance when the microcapsules added in were of 2%. Consequently, the effective service life of the coating can be extended about four times more than before [47].

Habib et al. (2019) synthesized multifunctional nanocomposite coatings by reinforcing an epoxy resin matrix with halloysite nanotubes (HNTs); they then added sodium nitrate as corrosion inhibitor and a self-healing agent (linseed oil (LO)) encapsulated with urea formaldehyde microcapsules (UFMCs). Then, they applied the epoxy nanocomposite coating on the polished mild steel using the doctor’s blade method. This smart nanocomposite coating could effectively resolve its mechanical damage through its self-healing ability. The efficiency of this coating in enhancing the corrosion resistance of the mild steel might be associated with its capability to quickly respond to external stimuli by a smooth release of the self-healing agent and inhibitor. Thus, this multifunctional epoxy nanocomposite coating system has great potential to be used as a corrosion protector in the oil and gas industries [48].

In the extrinsic self-healing mechanism, the self-healing functionality can also be acquired using a secondary phase of the coating material. In this case, the coating is in the form of polymer composite or polymer blend. It is crucial to understand the impacts of both the primary and secondary phase of the composite or blend on the healing performance of the coating system. The matrix properties also determine the efficiency of the self-healing functionality. For instance, Yuan et al. (2020) prepared self-healing coating systems using a blend of bio-based epoxy (made from diglycidyl ether of diphenolate esters (DGEDP) and thermoplastic polyurethane (TPU) prepolymers). The coating’s healing and mechanical properties were controlled by polymerization and induced phase separation morphology of the blend system. In particular, the molecular weight of the TPU prepolymer can be tuned to control the degree of phase separation of the blend. A thorough analysis on the relationships of the structure and properties was performed to prove this. They found out that the highly phase-separated morphology can be obtained by increasing the TPU prepolymer molecular weight. However, this phase-separated feature can only improve the mechanical performance, but not the healing functionality, of the resultant coating. Nevertheless, this study advocates the rational to control the morphology of the blend system in order to develop an efficient self-healing coating with targeted property profiles [49].

In summary, the polymer- and composite-based self-healing coatings were developed through the application of various methods and materials and are summarized in Table 2. However, further development is needed, like introducing novel healing methods, optimizing the engineering design, minimizing the cost, and accelerating the laboratory works into practical real-scale applications in several sectors and industries. Furthermore, it is crucial to develop durable self-healing polymeric coatings for long-term environmental exposure. Progress in this research area will pave the way for the development of multifunctional and smart self-healing polymers that are viable for several industrial sectors.

## 5. Self-Healing Polymers in Aerospace Application

Spacecrafts are expected to be tough enough to sustain in aerodynamic load. They are threatened by harsh space environmental factors, such as strong UV radiation, atomic oxygen, thermal cycles, space debris, vacuum, mechanical vibrations, and cosmic radiation [1]. The materials criteria in designing the aircraft structure must have appropriate mechanical properties with suitable damage tolerance under various conditions. Composite material has been introduced to withstand those high-impact conditions owing to their light weight, high mechanical properties, design flexibilities, excellent thermal resistance, and low cost [6,14]. The composite materials have been an increasing interest in the aerospace industry because of their properties, such as higher specific strength, fatigue resistance, better corrosion, and importantly, a great potential in not only applications, but also some engine parts. Therefore, designing the aircraft frames and engine with materials of improved mechanical properties can improve fuel efficiency, increase payload, and increase flight range, which all directly reduce the aircraft operating cost.

Common composite material use in spacecrafts are materials made of a polymer matrix reinforced with fiber, such as fiber-reinforced plastics (FRPs) reinforced with glass fibers (GFRPs), FRPs reinforced with carbon fiber polymers, and carbon-epoxy fiber-reinforced plastic (CFRP) polymer composites such as thermosetting epoxy reinforced with high-performance fibers [55]. Others matrix composites used in this application are ceramic matrix composites (CMCs), metal matrix composites (MMCs) and E-glass-epoxy composites (EGCs) [14]. However, composite material also endures deterioration due to impact load. The impact damage, which starts at the microscopic level from the formation of microscopic void, then is generated into a deep microcracking and delamination in structure, which reduces the structural integrity and leads to premature failure, as illustrated in Figure 8a–d [6].

In the previous approaches, damage was repaired by using resin patches, injection, and thermal plate methods. However, those approaches have several disadvantages like their inefficiency for invisible damages, needing damage monitoring, and the fact that they may not be applicable during the operation of the constructions. Moreover, the characteristics of the produced material are different from the individual component, as it has a short track record. Thus, the use of composites in aircraft components is limited due to their maintenance requirements. There is a recent study by Xiang et al. in which they fabricated a hybrid multiscale carbon fiber/epoxy composite with self-healing core shell nanofibers at interfaces via co-electrospinning. Co-electrospinning means the liquid solution of dicyclopentadiene (DCPD) as a healing agent was encapsulated into polyacrylonitrile (PAN) to form a core shell of DCPD/PAN nanofibers. The core shell DCPD/PAN is supposed to function to self-repair the interfacial damage in laminate composites, such as delamination [57]. The present interfacial toughening and co-electrospinning method was successfully characterized and these techniques are expected to be extremely beneficial in the development of high-strength, high-toughness, self-healing lightweight polymer matrix composites to be used in aerospace structures. However, there is still a challenge in this novelty, namely, interfacial fracture (delamination). Therefore, healing technology on the composite materials has more pros than cons. The aim is to reduce the damages or restore the functionality and lifetime of the deformed part, system, or device. Self-healing has proved to be remarkably efficient in the application of aerospace engineering. The healing concept application is further explained in the next subsection.

### 5.1. Self-Healing Mechanism in Aerospace

There are three groups of self-healing mechanisms in aerospace, which are com-posite materials, metal alloys for current conductors, and ceramic materials that are of direct interest to space technology. The first group is composite materials that show a major use in self-healing applications and which could be used as prime candidates for large structures and compartments, as well as for special (i.e., anti-corrosive) coatings. Besides that, the second group includes self-healing metal alloys, current conductors, and ceramic materials that can be used as a base for high-temperature elements of thrusters, long-living hard coatings, ceramic elements of electronic circuitry, and many other key systems. Figure 9 shows the classifications of materials used for the aerospace applications.

Similarly, the self-healing approach is based on extrinsic and intrinsic properties. For intrinsic properties, the self-healing forms from the presence of dynamic covalent bonds and the reversible formation of the cross-linker. Capacity of the free radicals is produced as a result of the mechanical damage that occurs to restructure covalent bonds. Meanwhile, for extrinsic properties, self-healing and adaptive capabilities are possible via special design approaches such as encapsulated healing agents, adhesion promoters, nanomotors, microencapsulated catalysts, and many others. The ability of a material to heal the damages in its structure relies on the type of material and function.

### 5.2. Self-Healing Polymeric Materials and Their Efficiency as Healing Agents in Aerospace

In such applications of self-healing materials, numerous studies have been carried out to access the efficiency of different self-healing polymers to fix the cracks in aerodynamic structures. Recent work in the area has involved self-healing composite materials, where the composite is made of epoxy vinyl ester, bismaleimide tetrafuran (2MEP4F), silicones like polydimethylsiloxane (PDMS), cyclopentadiene derivatives or cyanate ester. Raw polymer and diverse composites like E-glass fiber-reinforced composites (FRCs) and carbon FRCs were also investigated [14]. Polymers have a good molecular mobility, which explains why the majority of research is focused on polymers and their composites as self-healing materials [14].

Self-healing using carbon fiber-reinforced plastic (CFRP) polymer composites was designed in a major application of self-healing materials in aerospace by applying microcapsules loaded with healing materials to prevent delamination fracture of CFRP composites. The healing agent, a dicyclopentadiene-encapsulated microcapsule (d = 166 µm) was mixed with an epoxy resin by 20 wt%. The inter laminar fracture toughness of the specimens was recovered up to 40% and 80% at room temperature and 80 °C, respectively. Another method to recover from delamination damage was examined using a thermoplastic polymer matrix, the thermally responsive polyurethane combined with the Diels–Alder (DA) reaction achieved repeated healing of the delamination inside a carbon fiber composite with 85% and 75% healing efficiency throughout the first and second cycles, respectively [58].

Research on autonomic self-healing has also been reported on CFRP. A bespoke hollow glass fiber (HGF) manufacturing facility was used to produce fibers between 30 µm and 100 µm diameter with a hollowness of approximately 50%. These were embedded within either glass fiber-reinforced plastic (GFRP) or carbon fiber-reinforced plastic (CFRP) and infused with uncured resin to impart a self-healing functionality to a laminate. When damage occurs, some of these resin-filled fibers rupture, releasing the stored healing agent into a damaged site, thus initiating the recovery of the properties healing. This configuration achieved 97% of the undamaged state and 89% of the baseline laminate performance [59]. Coating is the most used method of application of self-healing products in aerostructures as it comprises polyvinyl ester matrix complemented by an adhesion promoter; thus, a dimethyldineodecanoate tin catalyst that is microencapsulated within polyurethane and a phase-separated poly(dimethylsiloxane) healing agent was used. The self-healing behavior of this system is enabled by the synthesis and encapsulation of Si [OSn(n-C_4_H_9_)_2_OOCCH_3_]_4_, which catalyzes the curing of PDMS [60].

Research also has reported on the application of an ethylene/methyl methacrylate (EMMA) copolymer as a healing agent for coating application in aerostructures. This polymer shows excellent self-healing ability and resistance to high-velocity impact. Others also use this coating as a thermal control coating which responds to UV radiation. Recent research by Zhu et al. (2019) introduced UV-responsive self-healing microcapsules for aerospace coatings. In this study, a UV-responsive microcapsule-based coating for in-orbit damage repair was developed. UV-responsive microcapsules have an inner polymeric shell that can be degraded rapidly by the outer, pure TiO_2_ shell under UV radiation, and they are produced by UV-initiated polymerization of Pickering emulsions and subsequently embedded into silicon resin matrices. When damaged, some microcapsules are ruptured as a result of external force, then the unbroken ones around the scratched areas are degraded by UV radiation, allowing the encapsulated healing agents to be released, and finally, to repair the cracks. Because of the dual-release mode, more healing agents are then released [1]. Coating has vital role in aerospace structures, and it involves the fuselages, wing, engine, cascade, etc. as a protective barrier. Table 3 tabulates the examples of self-healing polymeric materials used in aerospace applications.

The self-healing polymer has proven to be remarkably efficient in aerospace applications; however, self-healing polymer matrix composites are found more in aerospace applications compared to metal and ceramic composites because these composites can be manufactured easily and are less expensive. Besides that, self-healing concepts in metals and ceramics are relatively new, hence, it is more complex and expensive to manufacture them. Moreover, higher costs for testing and production are required for more investigation on their practical use. Overall, self-healing applications in aerospace structures are also restricted by some challenges such as the self-healing material must be continuously active and exposed under extreme environmental conditions.

## 6. Self-Healing Polymers in Medical Application

Hydrogels are an example of a three-dimensional-structure polymeric network that contains high water content and can be easily designed to mimic soft tissue compared to the metallic and ceramic biomaterials. Generally, self-healing hydrogels are one of the fundamental properties that allow for living tissue to recover their structures and restore their function in an automatic way after destruction [62]. The past decade has seen an increase in the usage of self-healable hydrogels as biomaterials in medical applications to replace brittle hydrogels because of their potential having the intrinsic ability of automatic damage repair without external interventions, and their long-term stability to be used in the therapy of physical defects and traumas. An ideal self-healing hydrogel material should have tunable structures for different applications and good rheological behavior, as well as being able to conduct intrinsically fast healing at the damage site under physiological conditions.

Recently, some of the self-healing hydrogels have been developed to have injectable ability [63,64] and conductivity [65] for autonomous self-recovery and minimally invasive implantation, which enables them to be widely used, especially in medical field. Injectable self-healing polymer hydrogels have been manufactured to be facilely delivered in vivo without surgical operation, and they can be injected as numerous solid fragments at a target position in order to maintain the activity of the embedded cells and drugs [66]. For instance, a review reported on injectable method using chitosan and polyethylene glycol(PEG) as derivative-based self-healing hydrogels with imine bond mechanism to functions for cell therapy, tumor therapy, and wound-healing applications [66]. They concluded that fast reactions of the imine bond mechanism provide a good effect on the hydrogel, as it can integrate after injection.

Additionally, conductive hydrogels are also one of the increasing and developed biomaterials that are being actively explored by scientists in the healthcare field, which are expected to provide an advanced self-healing material for implantable, wearable sensors and functional tissue-engineered scaffolds [67,68]. This conductive polymer provides electronic conductivity and ionic conductivity that is formed by a conjugated π system with the overlapped π orbitals through the polymer chain of polymerization [69]. This conductive hydrogel gives a new level of perspective in control over biomaterials when implanted into the human body. Additionally, in recent years the combination of electrically conductive hydrogels and 3D printing also has risen to attention, which can accurately produce the extracellular matrix structures and provide extra functions of hydrogel itself. For example, conductive hydrogels have been utilized in tissue engineering approaches and electrosensitive tissue, such as nervous or cardiac muscles [70], and nerves [71], as electrical conductivity is an important function to this tissue.

### 6.1. Self-Healing Mechanism in Medicals

The self-healing mechanism of hydrogels can be divided into two general methods, which are physically cross-linking (non-covalent interactions) and chemical cross-linking (reversible covalent interaction), as illustrated in Figure 10 [72,73]. Physically cross-linking mainly applies to hydrogen bonds, electrostatic interactions, hydrophobic interactions, and host–guest interactions. On the other hand, chemical cross-linking is mainly applied to Schiff base or imine bonds, Diels–Alder reactions, and disulfide bonds to form an insoluble network. The stability and mechanical properties of the self-healing hydrogels depends on the type and strength of the linkages that are used during the hydrogel synthesis. Therefore, it is important to better understand the self-healing mechanism as it can affect in the mechanical properties and design of hydrogels for biomedical applications, allowing materials to self-heal damages intrinsically and automatically. In this section, we focus on the type of mechanism that exists in designing the self-healing hydrogels.

Self-healing hydrogels can be formed through physical conditions such as hydrogen bonds, electrostatic reactions, hydrophobic interactions and host–guest interaction. These forms of physical cross-linking are less stable and more sensitive to environmental conditions compared to those of chemical cross-linking. Among other types of physical cross-linking, hydrogen bonding is one of the most used interactions that is formed between a hydrogen atom and another atom. Hydrogen bonds are bound to highly electronegative atoms such as Nitrogen (N), Oxygen (O) and Fluorine (F). Hydrogen bonding has been used to achieve temperature responsiveness of self-healing hydrogels [73]. For example, the hydrogen bonding via freezing method has been used to develop the polyvinyl alcohol-based self-healing hydrogels [74].

Moreover, electrostatic reaction or ionic bonding is formed from electrostatic attraction between charged ions. In this reaction, when the hydrogels are cut into two pieces, they are able to self-repair automatically in a short period of time [75]. This electrostatic reaction mostly has been used in self-healing conductive hydrogels. For example, a thermosensitive conductive hydrogel was developed in a study using chitosan and gold nanoparticles that act as electrical cues. These hydrogels demonstrated the novelty in cardiac tissue engineering applications, as they can provide thermo-responsive electro-conductive hydrogels as well as enhance the myocardial constructs [70].

Moreover, hydrophobic interactions occur because of the aggregation of hydrophobics domain in aqueous media because of the high interfacial tension to minimize the contact with water. Generally, these reactions’ associated structures are thixotropic that allow for the hydrophobic aggregate structures to reform and recover after being disturbed [72,76]. Additionally, host–guest interaction is a type of non-covalent bond that is formed from the two kinds of groups or molecular recognition of host and guest moieties. This interaction is based on inclusion complexation between macrocyclic hosts and smaller guest molecules [75]. In supramolecular host–guest interaction, there is one moiety called the guest that is physically inserted to another moiety called the host, and they are held together by a non-covalent bond [62].

Chemical cross-linking or dynamic covalent bonding is a chemical bond that can be reversibly broken, such as Schiff base reaction [77], Diels–Alder reaction, and disulfide bond, which is widely applied in the self-healing hydrogels. A dynamic covalent bond is stronger; however, it has a slower dynamic equilibrium compared to physical cross-linking. Schiff bases, commonly known as imine bonds, are a form of dynamic covalent bond between the aldehyde group and amine groups. The cross-linked network of these interactions can be achieved by condensation between polyamines and dialdehydes [78]. One of the most used o dialdehydes in self-healing hydrogel is the benzaldehyde-difunctionalized poly(ethylene) glycol through imine interactions [21]. Basically, this type of aldehyde is mixed with polymer and functionalized with amino groups, for instance, chitosan and its derivatives [66]. The preparation of self-healing hydrogels that involve both aliphatic and aromatic dialdehyde provides stability and also increases the final mechanical properties of the gels.

Furthermore, a Diels–Alder reaction can be dynamic under physiological conditions and have thermally reversible characteristics. However, this reaction is limited because the bonds need a high temperature and long period to cleave and reform for self-healing properties. In current studies reported by the authors of [79], a Diels–Alder reaction was developed to form self-healing hydrogels of cellulose nanocrystals-polyethylene glycol. The hydrogels did not only provide good mechanical properties, but also good self-healing properties and excellent recovery performance. In addition, disulfide bonds can be formed under low temperature. The bonds are formed by the oxidation of thiols in favorable conditions, for instance neutral, alkaline pHs and room temperature. The self-healing effect via the disulfide method has been observed by the authors of [80], who presented a self-healing polymeric based on disulfide exchange of 1,2-dithiolane and dithiols. The hydrogels can reform under natural conditions or in low alkaline pH and can be controlled by temperature.

### 6.2. Self-Healing Polymeric Materials and Their Efficiency as Healing Agents in Medical Fields

There has been increasing research effort and specialization in the development of self-healing polymer hydrogels in the medical field, particularly in tissue engineering [81], biomedicine [82], and wound-healing applications [83]. There has been a large array of development achieved in polymer hydrogel as a smart biomaterial due to its ability to mimic the native extracellular matrix (ECM), tunable properties, biocompatibility, biodegradability, and support for tissue regeneration. Numerous natural biomacromolecules and synthetic polymers, such as liposome [84,85], peptide [86], nucleic acid [87], and polysaccharide [88], contain self-healing hydrogels and have shown promising applications in cell culture, tissue engineering, drug delivery, and wound dressing. An overview of the type of polymer used based on self-healing hydrogels with various mechanisms, along with their applications in the biomedical field, is shown in Table 4.

The use of synthetic polymer polysaccharides, such as chitosan, cellulose, alginic acid, hyaluronic acid, and their derivatives, has been the study of increased research in tissue engineering applications. Polysaccharide is an attractive material due to its biodegradability, polyfunctionality, and biocompatibility with living tissue according to the desired application [92]. For instance, a self-healing hydrogel was manufactured using chitosan and PEG derivatives based on imine bonds due to its highly biocompatible nature. It is an important property, because the hydrogel maintains its functions and the viability of the encapsulated drugs, cells, and proteins [66]. According to the reviewed literature, it was concluded that chitosan and PEG derivatives have a low mechanical strength because of the dynamic nature of the imine bonds. This disadvantage limits their application in bone tissue engineering; however, the mechanical strength of the hydrogels is suitable for injecting hydrogels [66].

Furthermore, researchers also developed an injectable self-healing hydrogel with a tunable two-layers structure using chitosan (CHI) and phytic acid (PA) by two-step reaction. Firstly, the chitosan was oxidized to aldehyde-chitosan (ACHI) and fabricated by a Schiff base reaction to CHI/ACHI. Then, the hydrogel underwent an ionic bond with PA for the second layer to achieve the CHI/ACHI/PA hydrogel [89]. This combination of a two-layer hydrogel exhibits a good biocompatibility, excellent antibacterial properties, and therefore, makes it suitable for wound-dressing application. Moreover, a study reported by Jinhui [90] used carboxymethyl cellulose (CMC), acrylic acid (AA), and aluminum hexahydrate (AL) as a conductive hydrogel and also achieved a good result in strain and thermal sensitivity as wearable strain sensors. The conductive hydrogels exhibit human motion in precise real time with high sensitivity, according to the human body.

Furthermore, the most conductive polymers used as conductive hydrogels for biomedical applications are polypyrrole (PPY) and polyaniline (PAni) [69]. However, the hydrogels are limited due to their low mechanical properties. To overcome this limitation, researchers usually incorporate conducting nanomaterials together, for instance carbon nanotubes, carbon black, and metallic nanoparticles, via physical cross-linking or dynamic covalent mechanism. Another study by the authors of [68] reported mussel-inspired non-covalent interactions in designing self-healing conductive hydrogels that build up from interactions between polymerized acrylamide (PAM) and polydopamine (PDA) and partially reduced graphene oxide (GO), resulting in PDA-pGO-PAM hydrogels. The hydrogels achieved good conductivity, showed high sensitivity in electrical signals when implanted in deep tissues of rabbit, and served as in vivo electrodes/stimulators and detectors because of the well-dispersed reduction of GO.

Although the use and studies of self-healing polymers in medical fields are increasing and becoming more effective in the biomedical field, there are major challenges that restrict the applications, like high cost and a suitable mechanism. Therefore, this self-healing hydrogel needs to be addressed and well-studied before preclinical in vivo applications as well as other medical applications.

## 7. Self-Healing Polymers in Construction Application

Concrete has been the building block of the construction industry for a very long time, and its usage dates back to the ancient civilizations like the Roman Empire [93]. Concrete is still the first choice for the selection of construction material due to its unique properties such as high compressive strength, favorable fire resistance, ease of casting, and low cost compared to other construction materials. Alongside its many favorable characteristics, a major concern pertaining to concrete is that the relatively low tensile strength of concrete tends to make it vulnerable to cracking.

Problems like cracking and damage within concrete may arise not only from design errors, like the inadequate selection of materials used in the construction, but also from the effects of environmental conditions like freezing and thawing, carbonation, and alkali-silica reactions [93], as can be seen in Figure 11. Cracking from any reason in concrete provides access for aggressive constituents like chloride, carbonate, or sulphate to enter the concrete. These attacks may cause two outcomes: the deterioration of the concrete and the corrosion of the reinforcements.

Concrete deterioration induced by cracking has time and again proved to pose severe threats to the safety as well as the durability of concrete structures. The conventional solution for repair and rehabilitation utilizes human labor, which has consequences in the annual increase of rehabilitation work and costs. In addition, it contributes to the waste of already depleting resources. However, not all damages in concrete are at the macroscale. Certain damages subjected to concrete are at the microscale level, and hence, seem invisible, or rather, barely visible or inaccessible cracks. Reparation works for such cracking or damages may therefore prove to be expensive and require high labor operational works. Hence, a more sustainable solution for dealing with concrete cracking and damage is called for.

Microencapsulation of specific constituents, either solid or liquid, has gained the attention of many researchers these past few years for a diverse spectrum of applications. Inspired by the concept of biological regeneration of living tissue, microencapsulated self-healing cementitious constituents have recently gained popularity in the civil engineering field. The success of self-healing materials embedded in concrete structures may provide an alternative solution to rectify issues of cracking and damages in concrete.

### 7.1. Self-Healing Mechanism in Construction

Microcapsules are created using physical methods as a result of either a mechanical action or a physical process. Recent studies by Palin D et al. (2015) reported the use of marine bacteria-based agents for self-healing of marine concrete exposed to harsh sea water [95,96]. A. Kanellopoulos (2017) found that bacteria-based agents placed in synthesized gelatine/acacia gum microcapsules to encapsulate liquid sodium silicate for use in the production of self-healing concrete were discovered to be very stable when exposed to strong alkaline solutions that mimic exposure to concrete’s alkaline environment [97].

To apply the self-healing microcapsules into concrete, the microencapsulated capsules are first prepared. These mechanically triggered self-healing microcapsules and their catalysts are then incorporated into the cementitious mixture during the concrete mixing process. Once cured, the self-healing microcapsules are embedded into the concrete structure. The mechanism for self-healing microcapsules in concrete is shown in Figure 12. If subjected to concrete cracking, the embedded self-healing microcapsules tend to rupture, releasing the encapsulated healing agent. When the healing agent is exposed to the embedded catalyst, it polymerizes and works to seal and heal the cracks [98].

The success of the self-healing mechanism’s application to concrete depends on these factors: the immediate breakage of the embedded microcapsules, the internal flow of the healing agents into the crack locations, the reaction of the self-healing agents with the catalysts, and the duration and quality of curing [99]. Hence, proper selection of the healing agents and catalysts and the capsule materials is crucial for the self-healing mechanism in concrete to succeed.

### 7.2. Self-Healing Polymeric Materials and Their Efficiency as Healing Agents in Construction

The self-healing capacity of concrete specimens using non-ureolytic bacteria such as Bacillus Cohnii immobilized in expanded perlite (EP), was studied by J. Zhang et al. (2017) [100]. As time progressed, the crack widths in the concrete specimens with bacteria gradually decreased, and this occured especially for the carriers that were immobilized. For the controlled specimen, the cracks barely healed because of the further hydration of the unhydrated cement particles, whereas for the smaller crack widths, the precipitation of calcium carbonate crystals took place. Hence, this study further proves that when process assessment with microstructure analysis occurs, it makes it feasible for the EP particles to act as novel bacteria carriers on the process of healing cracks in concrete. Once these cracks appear in the concrete matrix, these cavities do not only provide enough cover and oxygen to the immobilized bacteria, but also allow for the bacteria to come in contact with enough water.

From the study by A. Griño Khushnood et al. (2020) [101], it was proven that the compressive strength of concrete can be further improved with Bacillus subtilis. The research also showed that the iron oxide nanosized particles (IONPs) were the most effective immobilizers, followed by siliceous and limestone particles, when it comes to the preservation of the bacteria until cracks are generated. As for composites, C. Wang et al. (2017) showed progress in synthesizing a type of oil absorbent microspheres (OAM). They further explained the effects of OAM on the characteristics of the cement stones, cement slurs, and on how the self-healing elucidates the cement microcracks. The preparation process was stable, and it also covered the surface of the microsphere with fumed silica shell. Pickering polymerization was used to prepare the fumed silica shell, while the self-healing cement using the oil absorbent microspheres was utilized. These properties ensured that the microspheres had a good hydrophilicity and were more compatible to cement slurry [102].

In L. LV et al. (2016), healing agents with poly(phenol–formaldehyde) PF resin were suggested to have a higher stability of cementitious properties and made it possible to trigger the resins from crack propagation. The in situ polymerization method was used to synthesize PF microcapsules that had a higher volume of healing agent. As the average size of microcapsules increased and the shell thickness decreased, and the mechanical force to trigger the microcapsules increased correspondingly. Smaller-sized synthesized PF microcapsules had higher chances of being triggered mechanically by the crack compared to the bigger-sized microcapsules [103].

Moving on to the preparation of the porous silica microsphere by Liang et al. (2019), a bio-inspired autolytic mineral self-healing method was proposed. Hexadecylamine and tetraethoxysilane (TEOS) in analytical grade were the non-ionic template and silicon source, respectively, and through the catalysis of ammonia, the porous silica microsphere was synthesized [104]. The autolytic self-healing mineral microsphere was prepared successfully, and it consisted of porous silica microsphere adsorbed with sodium silicate and polyvinyl pyrrolidone (PVP) coating film. The calcination method was used to increase the adsorption capacity of the silica microsphere, while the film thickness was kept the same with the use of proportional coating materials.

In addition to that, Xu and Wang (2018) proved that the protective carriers for ureolysis-based bacteria are of low-alkali cementitious materials with calcium sulphoaluminate cement and 20% silica fume. The compressive strength was retained and the amount of water increased by 130% and 50%, respectively, compared to the plain mortar. Water is important for the self-healing process of cementitious composites and it was found that the wet/dry cycles were the best curing option in mortar specimens that had encapsulated bacteria. Destructive methods were carried out to quantify the strength improvement of the bacteria, whereas non-destructive methods were carried out to validate the self-healing performance of bacteria in cementitious composites [105].

Furthermore, the studies on the cultivation of self-protected bio granules by Sonmez and Ersan (2019) proved that the activated compact denitrifying core, ACDC bio granules do not have any impact on the strength development and are compatible with hardened mortar [106]. In their experiment, bio granules with activated denitrifying core were used in tested bio-mortar mixtures as a microbial agent. To minimize the interaction between the mortar matrix and the core bacteria, and contribute to the bio granules’ compatibility with mortar, a CaCO_3_-Ca_3_(PO_4_)_2_ coating was done. Fresher mortar is compatible to ACDC bio granules, hence they do not interfere with their setting and workability if their dosage in the mortar mixture is around 3.60% *w*/*w* cement (i.e., 2.50% of bacteria in the form of ACDC *w*/*w* cement) or less.

Finally, the efficiency of the self-healing mechanism in concrete columns and slabs damaged with horizontal loads and impact loads were investigated by K. Van Tittelboom (2015) and Tran Diep [107]. Long and short glass tubes were used as capsules protected by a mortar layer to act against the impact from the aggregates during the manufacturing process. The capsules in the reinforcement did not break despite pouring the second layer of concrete during the casting. From the radiograph it was concluded that the capsules survived the casting process without any additional protection. The beams were casted in different layers with their side faces covered; thus, water intrusion in the concrete matrix was made by using the Karsten pipe method. In conclusion, various self-healing polymeric materials have been developed in construction applications and are summarized in Table 5.

## 8. Summary

The versatility of the polymeric materials allows for the usage of these materials as self-healing agents for wide spectrum of applications, ranging from hard to flexible applications. Various types of polymeric materials, such as polymer alone, polymer composites, and hydrogels, can be used as self-healing materials. This paper shows the wide range of examples for polymeric materials used in different applications. The behavior of the self-healing mechanism has shown significant improvements on the crack properties for both the extrinsic and intrinsic approaches. However, the extrinsic mechanism only permits one-time healing as compared to the intrinsic mechanism, which permits repetitive healings. Thus, the intrinsic self-healing mechanism is much more favorable, as it not only increases the life cycle of the materials, but also provide longer protection. However, there are still many challenges faced in order to develop an ideal self-healing system. More emerging polymeric materials and different self-healing mechanisms are expected to be explored and investigated for the future development of efficient self-healing materials.

## Figures and Tables

**Figure 1 polymers-13-01194-f001:**
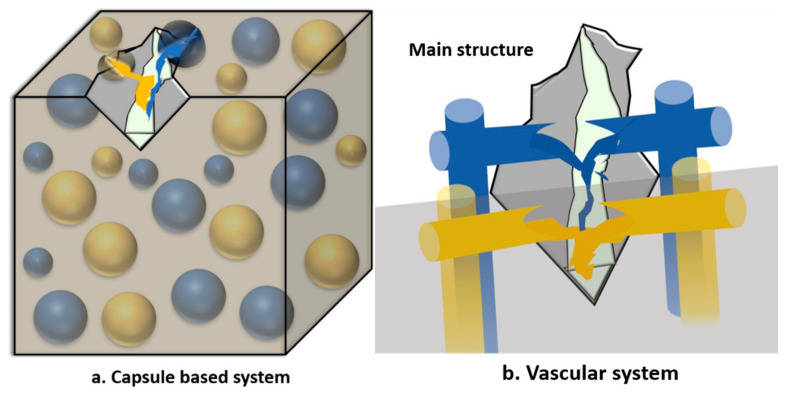
The extrinsic mechanism via (**a**) microcapsule and (**b**) vascular network.

**Figure 2 polymers-13-01194-f002:**
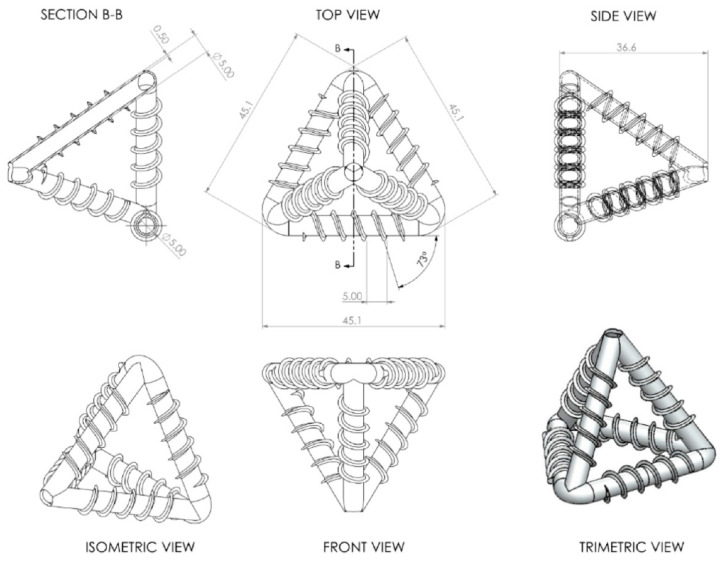
Mini-vascular networks (MVNs) of 3D-printed tetrahedral units (TETs) obtained from the authors of [25].

**Figure 3 polymers-13-01194-f003:**
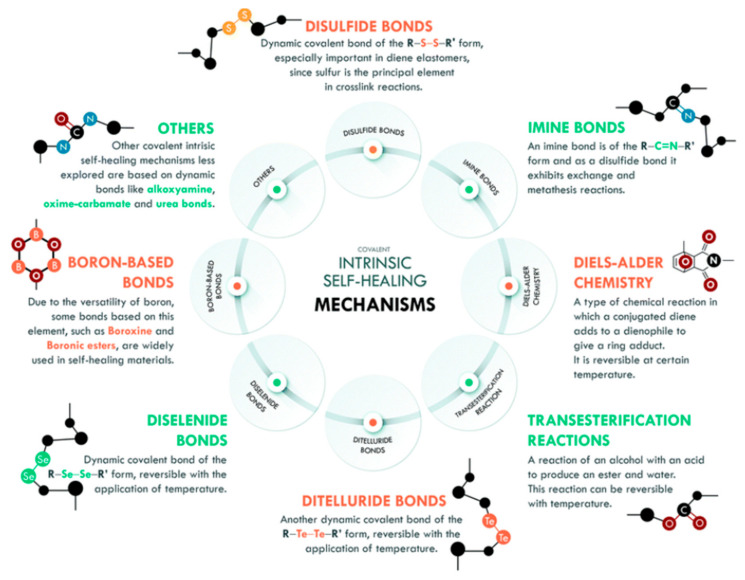
Covalent intrinsic mechanism obtained from the authors of [28].

**Figure 4 polymers-13-01194-f004:**
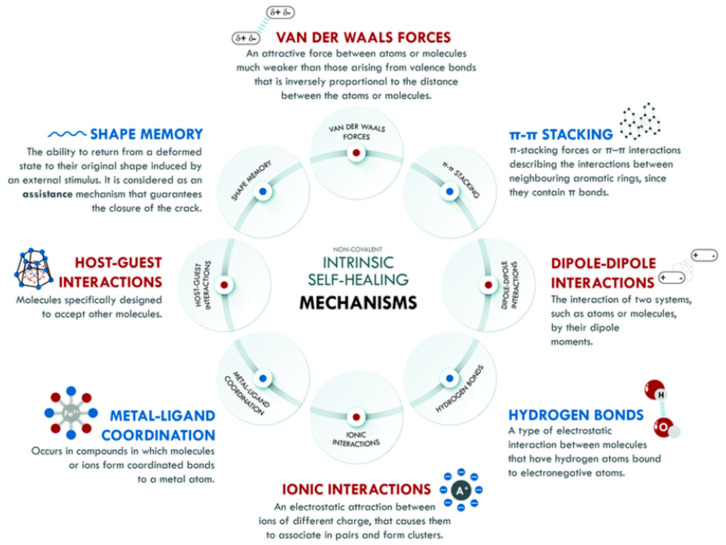
Non-covalent intrinsic mechanism obtained from the authors of [28].

**Figure 5 polymers-13-01194-f005:**
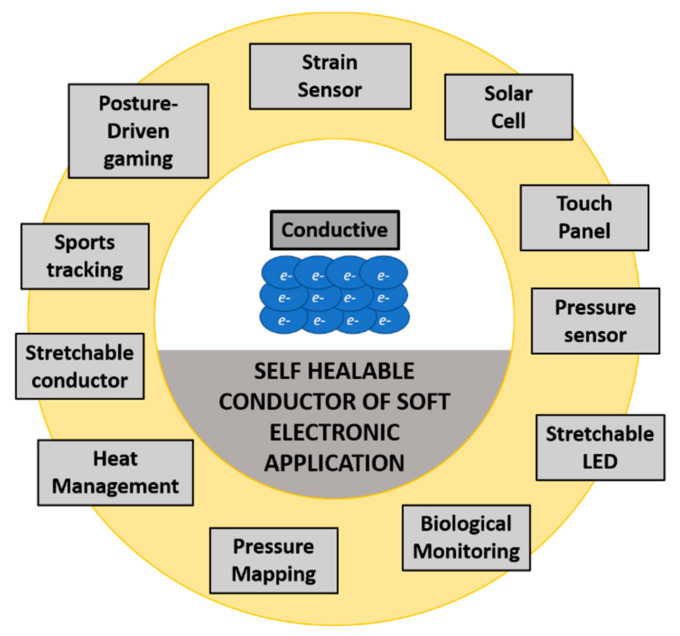
Soft electronic applications.

**Figure 6 polymers-13-01194-f006:**
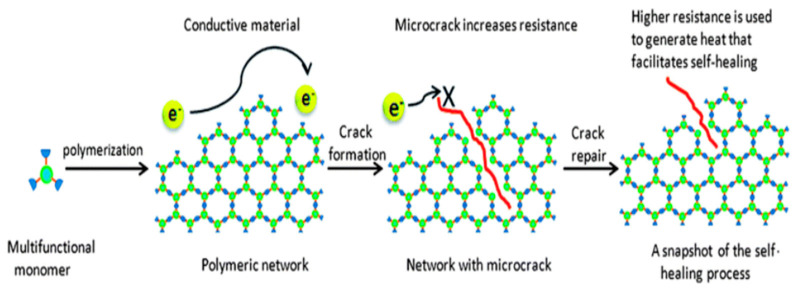
Fundamentals of electrically conductive self-healing materials. Electrical resistance increases upon formation of a microcrack, as the total number of electron percolation pathways decreases. As the microcrack is the source of the resistance increase, the generation of heat is localized at the fracture point according to the authors of [37].

**Figure 7 polymers-13-01194-f007:**
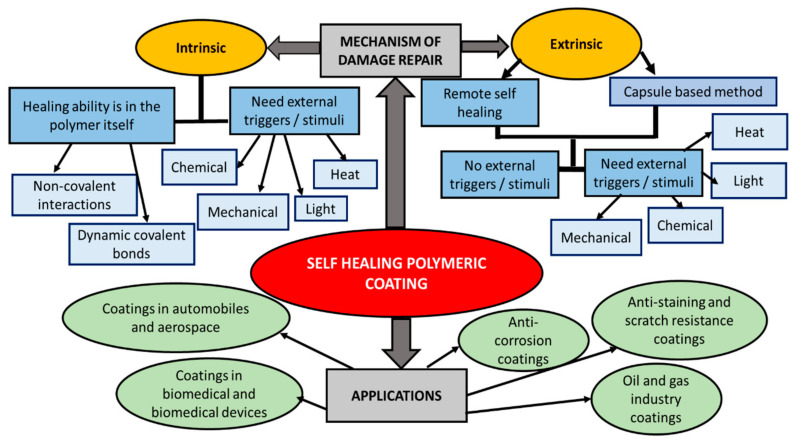
Self-healing polymeric coatings, mechanisms, and applications.

**Figure 8 polymers-13-01194-f008:**
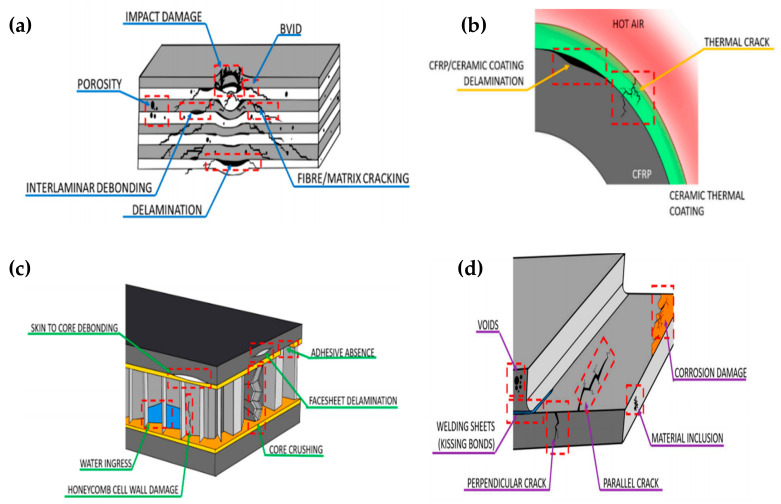
Type of damages in aerospace crafts: (**a**) jet engine turbine blades, (**b**) honeycomb panels, (**c**) metallic aircraft, and (**d**) spacecraft components. Images obtained from Reference [56].

**Figure 9 polymers-13-01194-f009:**
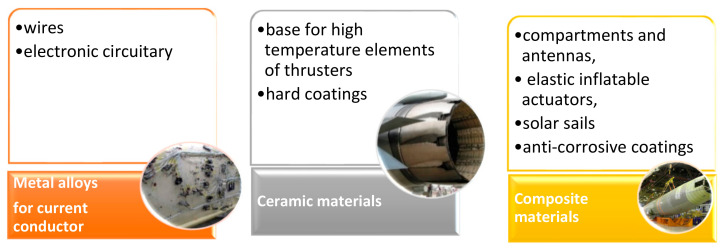
Classification of self-healing materials in spacecraft applications.

**Figure 10 polymers-13-01194-f010:**
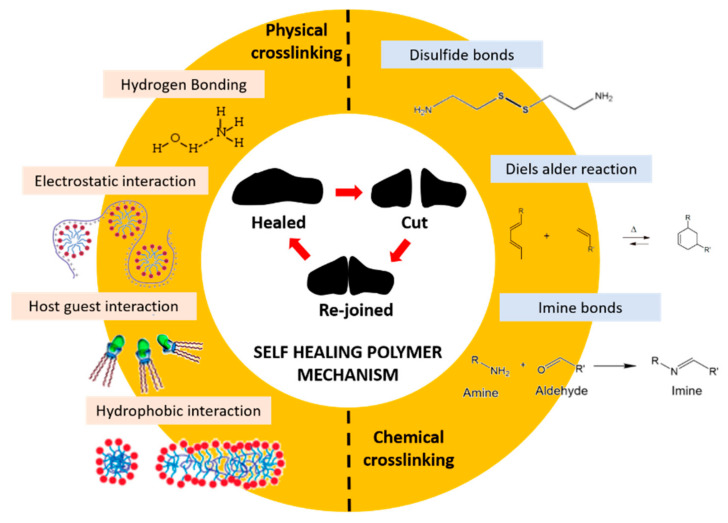
A summary of physical cross-linking and chemical cross-linking use in the self-healing polymer mechanism.

**Figure 11 polymers-13-01194-f011:**
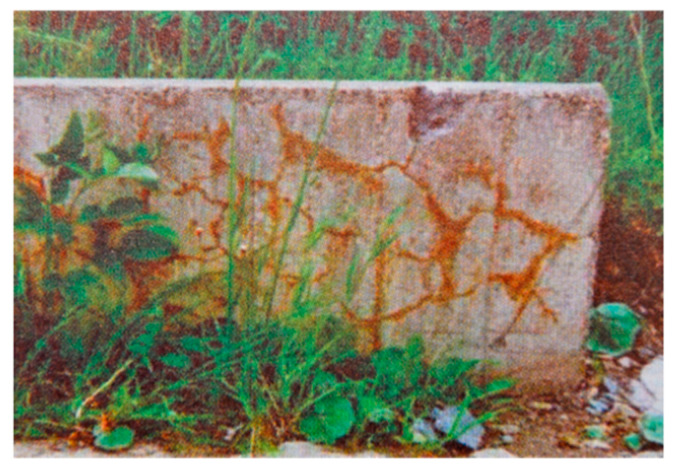
Map cracking due to alkali silica reaction, ASR deterioration on a concrete wall [94].

**Figure 12 polymers-13-01194-f012:**
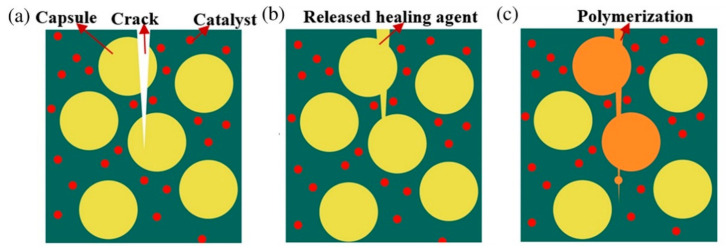
Self-healing schematic concept for cementitious materials that are (**a**) crack forming, (**b**) healing agent releasing, and in (**c**) polymerization with catalysts [98]. Images obtained with permission from C. Xue, W. Li, J. Li, V.W.Y. Tam, and G. Ye, “A review study on encapsulation-based self-healing for cementitious materials,” Struct. Concr., vol. 20, no. 1, pp. 198–212, 2019.

**Table 1 polymers-13-01194-t001:** Self-healing polymers in electronic applications.

Application	Polymer Use	Interaction /Mechanism	Ref
Smart batteries and supercapacitors	Conductive polymer gels (CPGs)	Reversible chemical bonds or specific interactions (ligand–metal bonding and hydrogen bonding)	[38]
Smart Batteries and supercapacitors	Polyvinylidene fluoride (PVDF), sodium alginate (SA), carboxymethyl cellulose (CMC), and polyacrylic acid (PAA)	Nil	[38]
Stretchable electronic device	Poly(acrylic acid) (PAA) and reduced graphene oxide (rGO)	Based on intermolecular networks covalent (ester bonds) and non-covalent (hydrogen bonding)	[35]
Stretchable electronic device	Poly (vinylidene fluoride-co-hexafluoropropylene), poly(3,4-ethylenedioxythiophene) polystyrene sulfonate (PEDOT:PSS), low-density polyethylene (LDPE), cross-linked polyethylene (XLPE), polypropylene (PP), polyimide (PI), epoxy, polydimethylsiloxane (PDMS), and polyvinylidene fluoride (PVDF)	Interacted with the ionic liquid through strong ion–dipole interaction	[32,33]
Stretchable electronic device	Poly(3,4-ethylenedioxythiophene): poly(styrene sulfonate) (PEDOT:PSS), polyaniline (PANi), polypyrrole (PPy)) as a healing agent, and polypyrrole and ferric ions	Reversible Diels–Alder (DA) reaction	[36]

**Table 2 polymers-13-01194-t002:** Self-healing polymers in coating applications.

Application	Types of Polymer/Composite	Interaction/Mechanism	Ref
General anti-corrosion coating	Microcapsules of urea formaldehyde as the wall material and epoxy resin (E-51) as the core material.	Extrinsic mechanism through capsule-based system (microencapsulation)	[47]
Surface coating for carbon steel.	Superabsorbent polymer (SAP), which is a modified cross-linked polyacrylate	The microfibers of a superabsorbent polymer (SAP) act as a multilayer polymer coating, which can be applied to carbon steel to promote corrosion inhibition.	[50]
Coating for seawater application	Isophorone diisocyanate (IPDI)	Alkyd varnish coating (AVC), with isophorone diisocyanate (IPDI) microcapsules as functional additives using a water trigger mechanism in seawater	[51]
Coating for biomedical magnesium alloys	Silk fibroin/phytic acid composite	Active, bio-corrosion-responsive self-repairing capacity through pH stimuli-responsiveness and osteogenic activity	[52]
Long-term active corrosion protection of aerospace grade aluminum alloy	Water-responsive inhibiting nanonetworks based on polyvinyl alcohol (PVA) electrospun nanofiber mats.	Based on the formation of low-density and/or humidity-responsive inhibitor interconnections in the coating, what are called inhibiting nanonetworks.	[53]
Coating for mild steel in oil and gas industries	Urea formaldehyde microcapsules encapsulate linseed oil (LO) self-healing agent	Extrinsic mechanism through capsule-based system (microencapsulation)	[51]
Long-term and durable coating for outdoor environment (sunlight-exposed area)	Polydimethylsiloxane (PDMS) as template and epoxy-based shape memory polymer (SMP) as superhydrophobic surface.	Self-healing superhydrophobic coating based on dual actions by the corrosion inhibitor benzotriazole (BTA) and an epoxy-based shape memory polymer (SMP)	[54]

**Table 3 polymers-13-01194-t003:** Self-healing polymer composites in aerospace applications.

Type of Polymer	Application	Interaction/Mechanism	Method	Ref
Dicyclopentadiene-encapsulated microcapsule mixed with an epoxy resin by 20 wt%	Carbon fiber-reinforced plastic (CFRP) polymer composites	-	Microcapsule	[58]
Polyurethane	Carbon fiber composite	Diels–Alder (DA) reaction	intrinsic	[58]
Hollow glass fibers (HGF) two-part epoxy healing agent (Cytec Cycom 823)	Carbon fiber-reinforced plastic (CFRP) polymer composites	Nil	Autonomous/fibers between 30 µm and 100 µm diameter	[59]
Phase-separated poly(dimethylsiloxane) (PDMS), epoxy vinyl ester matrix (adhesion promoter), dimethyldineodecanoate tin (catalyst) encapsulate in poyurethane	-	Synthesis and encapsulation of Si [OSn(n-C4H9)2OOCCH3]4 that catalyzes the curing of PDMS	Microcapsule	[6]
Epoxy silicon with TiO_2_ nanoparticles	Coating	UV-initiated polymerization when exposed to UV radiation, the first outflows are initiated and solidified, while the unbroken microcapsules around the cracks are degraded by the outer TiO_2_ shell simultaneously	Microcapsule	[1]
2,4,6-tris [(dimethylamino) methyl], initiator Phenol with Grubb catalyst	-	ring-opening metathesis polymerization (ROMP)	Microcapsule	[19]
Dicyclopentadiene (DCPD) and isophorene diisocyanate	Fiber-reinforced polymer composites	Encapsulated self-healing materials inside polymer fibers, such as polyacrylonitrile (PAN) and amorphous turbostratic carbon nanotubes, via co-electrospinning, emulsion electrospinning, and emulsion solution blowing	Encapsulation	[55]
Core shell nanofiber system of polyacrylonitrile (PAN) shell via co-axial electrospinning	Polymer matrix composites	Dual components of a self-healing epoxy system consisting of low-viscosity epoxy resin and an amine-based curing agent were separately encapsulated in PAN	Encapsulation	[61]

**Table 4 polymers-13-01194-t004:** Self-healing hydrogels in medical applications.

Polymer Self-Healing Hydrogels	Self-Healing Mechanism	Healing Efficiency and Conditions	Application	Ref
CHI/ACHI/PA	Schiff base reaction and ionic bond	Enhanced mechanical properties, modulus increased reaching 5000 pa	Adhesion and antibacterial properties	[89]
PAA-CMC-Al^3+^	Dynamic non-covalent interaction	Excellent thermal sensitivity, increased mechanical properties, healing efficiency: 96.3% within 60 min	Wearable skin sensors	[90]
PDA-pGO-PAM conductive hydrogels	Dynamic non-covalent interaction	Good conductivity, high sensitivity	In vivo electrodes and detector	[68]
Adhesive liposome (A-lip) with PEG	Electrostatic reaction	Maintain adhesive properties, drug delivery, reduce the risk of postoperative infection, minimally invasive injection	Bone reconstruction	[85]
Chitosan-gold nano particles (CS-GNP)	Electrostatic reaction	Viability, enhanced myocardial construct, electro-conductive hydrogels	Cardiac tissue engineering	[70]
Dialdehyde-Hyaluronic acid (AHA)/cystamine dihydrochloride (Cys) AHA/CYS	Schiff base reaction	Healing efficiency 100% based on the stress within 10 min, fast gel formation, improved mechanical properties, pH responsiveness,	Tissue regeneration	[77]
Acrylamide modified β-chitin/alginate dialdehyde	Schiff base reaction and hydrogen bond	Exhibit excellent biodegradability, biocompatibility, and injectability	Wound healing	[83]
Adipic dihydrazone-grafted carboxyethyl chitin (CECT-ADH)/di-benzaldehyde-terminated poly (ethylene glycol) (PEG-DA)	Dynamic covalent bond Acyl hydrazone cross-linking	Excellent self-healing capacity with 95% healing efficiency, support proliferation and multipotent differentiations of rat bone marrow-derived stem cells	Injectable delivery vehicle of therapeutic drugs or cells for tissue regenerative medicine	[91]

**Table 5 polymers-13-01194-t005:** Self-healing materials in construction applications.

Self-Healing Agent	Application	Findings	Ref
Non-ureolytic bacteria, i.e., Bacillus cohnii, immobilized in EP	Self-healing method for cementitious materials	The crack widths of bacteria-based concrete specimens, particularly those immobilized in the carriers, were found to gradually decrease over time. The cracks of the control specimen were barely healed.	[100]
Self-healing cement using oil absorbent microspheres with a fumed silica shell, prepared by Pickering polymerization.	Self-healing method for cementitious materials	The microsphere had good hydrophilicity, which made it have good compatibility with cement slurry.	[102]
Clinker/PVP autolytic microsphere	Self-healing method for cementitious materials	Experimental results showed that the compressive strength recovery of cement paste with a 30% microsphere was 54% higher than ordinary cement paste specimens. The damage degree of the specimen was also decreased by adding the autolytic microsphere.	[108]
Phenol, formaldehyde (37%), sodium hydroxide (NaOH), hydrochloric acid (HCl), poly (acrylic acid sodium salt) (PAA-Na) (Mw = 1200), and liquid dicyclopentadiene (DCPD) PF microcapsules.	Self-healing method for cementitious materials	The result demonstrated that smaller-sized synthesized PF microcapsules tend to be more mechanically triggered by the crack.	[103]
Bio-inspired autolytic mineral microsphere: hexadecylamine and tetraethoxysilane (TEOS) and ammonia catalyst	Self-healing method for cementitious materials	Adsorption amount increases by 39%, leading to better healing potential.	[104]
Combination of microcapsules, bacteria, shape memory polymers, and flow networks	Self-healing method for cementitious materials	The compromise between the shrinkage stress generated and the number of tendons embedded into the concrete showed that this technique is feasible at this larger scale. The installation of the flow networks in these full-scale panels demonstrated that it was feasible to repeatedly flush the healing agent through the cracks in the panels.	[109]
Bio-influenced self-healing mechanism	Self-healing method for cementitious materials	In future, microencapsulated technology could be the one emerging method to evaluate the performance of the bacteria and detect real-time cracks inside the concrete matrix. However, there are still existing issues and challenges associated with the use of bacteria in the field of the construction industry.	[101]
Synthesis of gelatine/acacia gum microcapsules to envelope liquid sodium silicate	Self-healing method for cementitious materials	Microcapsules were found to be very stable when exposed to strong alkaline solutions that mimic exposure to concrete’s alkaline environment. Thermogravimetric tests revealed that the produced microcapsules were very stable up to 190 °C.	[97]
Encapsulation materials containing polyurethane	Self-healing method for cementitious materials	Evaluation of the crack repair efficiency by performing water permeability tests showed some improvement in water tightness due to self-healing, but the water ingress into the cracks was not completely prevented.	[107]
Cultivation of self-protected bio granules with an activated denitrifying core as the microbial agent.	Self-healing method for cementitious materials	ACDC bio granules are compatible with fresh mortar and do not interfere with their setting and workability if their dosage in the mortar mixture is around 3.60% *w*/*w* cement (i.e., 2.50% of bacteria in the form of ACDC *w*/*w* cement) or less. ACDC bio granules have no influence on strength development and they are compatible with the hardened mortar.	[106]

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
