# Peer review of "The Versatility of Polymeric Materials as Self-Healing Agents for Various Types of Applications: A Review"

_polymers, 2021, doi:10.3390/polym13081194_

Round 1

Reviewer 1 Report

  1. Evidence (references) for the market data described in the introduction is required.
  2. Some figures resolutions was low quality. Please recheck figure content. (ex. Figure 5)
  3. Figure 7 show, self-healing polymeric coating mechanisms and application. Actually, it is not appropriate to distinguish between autonomous and non-autonomous mechanisms for repairing damage. Because of most of the currently reported intrinsic and extrinsic self-healing also require extra energy (stimulation) to restore damage. Therefore, the correction of Figure 7 is required.

(ex.  Intrinsic self-healing  ß Mechanism of Damage repair à Extrinsic self-healing )

Author Response

Thank you for giving us the opportunity to submit a revised draft of manuscript titled The Versatility of Polymeric Materials as Self-Healing Agent for Various Types of Applications: A Review to Polymers. We appreciate the time and effort that you have dedicated to providing your valuable and constructive feedback in this manuscript. We are grateful to the reviewers for their insightful comments on this paper. We have been able to incorporate changes to reflect most of the suggestions provided by the reviewers. We have highlighted the changes within the manuscript.

Reviewer 2 Report

The manuscript polymers-1154046 entitled “The Versatility of Polymeric Materials as Self-Healing Agent for Various Types of Applications: A Review” reviews the recent advances for the development of self-healing polymeric materials. The subject is interesting but the manuscript needs major revisions before it can be recommended for publication in Polymers.

Comments:

  1. The manuscript needs a significant rephrase in the sentences. Line 116 up to the end of the manuscript must be rephrased. All the sentences are similar to the published articles.
  2. In the last paragraph of the introduction, the authors should list the reviews already published in this area and state what is special about this review.
  3. References must be edited according to journal instruction. For example, line 123 [6][7][8][9] must be changed into [6-9].
  4. Figure 2 should be redrawn by the authors or the quality of the figure must be improved.
  5. Section 5. The recent works by Yarin’s group and Ramakrishna’s group on the development of self-healing laminated composites should be reviewed and the methods should be discussed. Please see the following articles.

a) Wu, X.-F.; Rahman, A.; Zhou, Z.; Pelot, D.D.; Sinha-Ray, S.; Chen, B.; Payne, S.; Yarin, A.L. Electrospinning core-shell nanofibers for interfacial toughening and self-healing of carbon-fiber/epoxy composites. Journal of Applied Polymer Science 2013, 129, 1383-1393, doi:10.1002/app.38838.

b) Sinha-Ray, S.; Pelot, D.D.; Zhou, Z.P.; Rahman, A.; Wu, X.F.; Yarin, A.L. Encapsulation of self-healing materials by coelectrospinning, emulsion electrospinning, solution blowing and intercalation. Journal of Materials Chemistry 2012, 22, 9138-9146, doi:10.1039/C2JM15696B.

c) Neisiany, R.E.; Khorasani, S.N.; Kong Yoong Lee, J.; Ramakrishna, S. Encapsulation of epoxy and amine curing agent in PAN nanofibers by coaxial electrospinning for self-healing purposes. RSC Advances 2016, 6, 70056-70063, doi:10.1039/C6RA06434E.

d) Neisiany, R.E.; Lee, J.K.Y.; Khorasani, S.N.; Ramakrishna, S. Towards the development of self-healing carbon/epoxy composites with improved potential provided by efficient encapsulation of healing agents in core-shell nanofibers. Polymer Testing 2017, 62, 79-87, doi:https://doi.org/10.1016/j.polymertesting.2017.06.016.

e) Neisiany, R.E.; Lee, J.K.Y.; Khorasani, S.N.; Ramakrishna, S. Self-healing and interfacially toughened carbon fibre-epoxy composites based on electrospun core–shell nanofibres. Journal of Applied Polymer Science 2017, 134, 44956, doi:10.1002/app.44956.

Author Response

Thank you for giving us the opportunity to submit a revised draft of manuscript titled The Versatility of Polymeric Materials as Self-Healing Agent for Various Types of Applications: A Review to Polymers. We appreciate the time and effort that you have dedicated to providing your valuable and constructive feedback in this manuscript. We are grateful to the reviewers for their insightful comments on this paper. We have been able to incorporate changes to reflect most of the suggestions provided by the reviewers. We have highlighted the changes within the manuscript.

Please see attachment for the point-to-point correction. Thank you.

Round 2

Reviewer 2 Report

The manuscript can be recommended for publication.